# Long-Term Application of a Synbiotic Chitosan and *Acinetobacter* KU011TH Mixture on the Growth Performance, Health Status, and Disease Resistance of Hybrid Catfish (*Clarias gariepinus* × *C. macrocephalus*) during Winter

**DOI:** 10.3390/microorganisms11071807

**Published:** 2023-07-14

**Authors:** Pisey Say, Sukkrit Nimikul, Anurak Bunnoy, Uthairat Na-Nakorn, Prapansak Srisapoome

**Affiliations:** 1Laboratory of Aquatic Animal Health Management, Department of Aquaculture, Faculty of Fisheries, Kasetsart University, Chatuchak, Bangkok 10900, Thailand; say.p@ku.th (P.S.); anurak.bun@ku.ac.th (A.B.); 2Center of Excellence in Aquatic Animal Health Management, Department of Aquaculture, Faculty of Fisheries, Kasetsart University, Chatuchak, Bangkok 10900, Thailand; 3Department of Aquaculture, Faculty of Fisheries, Kasetsart University, Chatuchak, Bangkok 10900, Thailand; ffisskn@ku.ac.th; 4Laboratory of Aquatic Animal Genetics, Department of Aquaculture, Faculty of Fisheries, Kasetsart University, Chatuchak, Bangkok 10900, Thailand; ffisurn@ku.ac.th; 5Academy of Science, The Royal Society of Thailand, Bangkok 10300, Thailand

**Keywords:** *Acinetobacter* KU011TH, *Aeromonas hydrophila*, chitosan, *Clarias* catfish, disease resistance, hybrid catfish

## Abstract

The effects of potential synbiotic chitosan and *Acinetobacter* KU011TH mixture on growth performance, immune response, and *A. hydrophila* resistance were investigated for the first time. The control group was fed a basal diet (A), and group B was given the formula B diet that was supplemented with chitosan at 20 mL/kg diet via top dressing. The other synbiotic groups, C, D, and E, were top-dressed with the target probiotics at 1 × 10^8^, 1 × 10^9^, and 1 × 10^10^ CFU/kg diet, respectively, and coated with the same concentration of chitosan. Fish were continuously fed the five different feeds for 16 weeks during winter. At the end of the trial, the growth parameters of the test groups did not significantly differ from those of the control (*p* > 0.05). All the symbiotic-chitosan treatments significantly increased various hematological and serum immune parameters. Moreover, the expression levels of immune-related genes were strongly elevated in the head kidney and spleen, whereas upregulated expression was observed in the liver and whole blood (*p* < 0.05). Survival analysis indicated that fish in groups B and C showed significantly higher survival (84.33 ± 2.21 and 79.50 ± 6.34%) than those in groups A, D and E (55.33 ± 8.82%–74.00 ± 6.50) (*p* < 0.05) after injection with *A. hydrophila* for 14 days.

## 1. Introduction

Catfish of the genus *Clarias* are distributed in freshwater habitats across Africa and Asia. At least three species of this genus, African catfish (*Clarias gariepinus*) (Burchell, 1822), bighead catfish (*C. macrocephalus*) (Günther, 1864), and *C. batrachus*, have been intensively cultured [1]. These species have the ability to tolerate different water conditions and can stay out of water for several hours and tolerate low oxygen concentrations as they have accessory organs that enable them to breathe atmospheric air [2,3]. Their production has been practiced with artificial breeding to date. Breeding has led to the development of production systems and hatchery management techniques that allow eggs to reach a good quality [4].

Hybrid catfish (*Clarias macrocephalus* × *C. gariepinus*) are one of the most popular cultured freshwater species in Southeast Asia. These fish constitute a large proportion of cultured catfish. The annual total production of catfish in the Kingdom of Thailand was approximately 102,375 tons in 2019, 95% of which came from hybrid catfish [1]. Moreover, this hybrid catfish exhibits high production and acceptable flesh quality. Specifically, hybrid catfish have shown many advantages over both bighead and African catfish, such as a fast growth rate, decent feed conversion, resistance to infectious pathogens, and tolerance to various environmental culture conditions. Consequently, these fish have a high market value and are exported to neighboring countries and overseas as part of Thailand’s fish and fishery products [5,6].

In southeast Asia, the growth rate of all aquatic animals and cultured fish (including catfish), as poikilothermic animals, is seriously affected by low temperatures in winter. During this season, some key water parameters, especially alkalinity, are also at lower than optimal levels, which is caused by monsoon rain and the high volume of seepage water from heavy rain. Accordingly, these conditions have led to a nonideal growth rate and harmful diseases, notably bacterial infections. A severe pathogen, *Aeromonas hydrophila*, is the most economically critical candidate and has led to tremendous concern, leading to growth retardation and mass mortality during this period [7,8].

To overcome these problems, much attention has been given to feed additives, growth promoters, beneficial microbes, and immunostimulants, which have been explored to improve health status and production in aquatic animals while promoting ec-friendly aquaculture systems. Of these, prebiotics are compounds in food that effectively foster the growth or activity of beneficial microorganisms, probiotics are live microorganisms that provide health benefits when consumed or applied to the body, and synbiotics are mixtures of probiotics (beneficial microorganisms) and prebiotics (nondigestible fibers that help these microorganisms grow and fully function). Synbiotics are considered nonspecific immunostimulants that play an effective role in the host body to modulate the immune system and enhance resistance against diseases in almost all circumstances [8,9]. Their application was linked to proven effects on the gut microbiota, resulting in improved health status via two main mechanisms: a nutritional effect and a health effect. The nutritional effect reduces the production of toxic substances and stimulates enzyme activities, while the health effect increases pathogen colonization resistance and enhances the immune response [10,11,12,13]. Additionally, a food safety assessment system, namely, the Qualified Presumption of Safety (QPS), which is launched by the European Food Safety Authority (EFSA), also involves safety criteria for bacterial supplements, including safe usage history and absence of antibiotic resistance risk [14]. The application of synbiotic approaches is also helpful and provides promising benefits for the aquaculture of various aquatic animals [15,16]. Assessment of the potential applications of synbiotics in the aquaculture industry in various regions requires further investigation and development.

Recently, various probiotics and prebiotics have been tested in various kinds of aqua-cultured fish, but the applications of synbiotics have not been widely studied. These synbiotics include *Enterococcus faecalis*/mannanoligosaccharide (MOS) or polyhydroxybutyrate acid (PHB), *Bacillus clausii*/MOS or fructooligosaccharide (FOS), *Bacillus subtilis*/FOS, and *Bacillus subtilis*/chitosan, which were tested in rainbow trout (*Oncorhynchus mykiss*), Japanese flounder (*Paralichthys olivaceus*), cobia (*Rachycentron canadum*), and large yellow croaker, respectively (reviewed in [12]). In *Clarias* catfish, a few probiotic strains, including *Lactobacillus* sp., *Lactobacillus plantarum*, *Pseudomonas fluorescens*, *Bacillus subtilis*, and *Parkia biglobosa*, were shown to improve growth, water quality, and health status [17,18,19]. Furthermore, potential prebiotics, probiotics, and synbiotics have not been extensively studied in hybrid *C. gariepinus* × *C. macrocephalus* catfish, in which *Bacillus* W120 and *Enterococcus hirae* were identified as having evident probiotic effects [20,21].

Bacteria of the genus *Acinetobacter* are Gram-negative, nonmotile, catalase-positive, oxidase-negative, and coccobacillus-like aerobes that survive under a wide range of environmental conditions. A recent study of the genus *Acinetobacter* recorded 61 distinct species that were isolated from a variety of sources, including human clinical specimens, invertebrate and vertebrate animals, vegetables, soil, and water [22,23]. Although research on the genus *Acinetobacter* has focused on species that have impacts on human health, particularly *A. baumannii* [24], some species were also found to be pathogenic to some fish [25,26,27,28]. The beneficial effects of other strains on their hosts have been proven in a limited number of studies. Only *Acinetobacter* haemolyticus was found to strongly inhibit *Tenacibaculum maritimum* in farmed turbot (*Scophthamus maximus*), but application trials of this species were not further conducted [29].

Recently, *Acinetobacter* KU011TH was successfully isolated from the skin mucus of a healthy bighead catfish. This strain was demonstrated to be an effective probiotic, positively affecting both the growth and health performance of bighead catfish [9,30]. To expand its potential applications, a specific prebiotic-like compound such as chitosan was targeted to increase its ability in enhancing the growth and immune responses of hybrid *Clarias* catfish. Concurrently, a commercial product derived from chitin or chitosan sourced from a range of aquatic organisms is being utilized for diverse purposes. Chitosan and chitooligosaccharide (COS) are derived from various raw materials, including terrestrial crustaceans and mushrooms. This biopolymer is biodegradable and possesses many beneficial properties, such as biocompatibility, nontoxicity, and antimicrobial, hemostatic, anti-inflammatory, antioxidant, and adsorption enhancement effects [30,31]. Recently, in an aquaculture system, chitosan was experimentally applied, and its feed supplementation optimally improved probiotic potency under in vivo gastrointestinal conditions and consequently resulted in promising non-plant-delivered prebiotics with multiple biological activities [32,33] in some fish species, including brown trout (*Salmo trutta fario*) [34], hybrid grouper (*Epinephelus fuscoguttatus* ♀ × *E. lanceolatus* ♂) [35], silver barb (*Barbonymus gonionotus*) [36], and Nile tilapia (*Oreochromis niloticus* (L.)) [37]. However, its prebiotic effects on *Acinetobacter* KU011TH and the synbiotic effects of these components on *Clarias* catfish or its hybrid have never been investigated.

Based on the above knowledge, this study aimed to investigate the effects of chitosan and *Acinetobacter* KU011TH application on the growth performance of hybrid catfish during winter after long-term application for 16 weeks. Additionally, the health status, immune responses, and disease resistance against *A. hydrophila* of hybrid catfish were investigated at the end of the trials. The results and findings of the current research could be further applied to improve the growth rate and disease resistance during critical winter periods in fish cultivation. This novel knowledge will be an additional strategy for catfish cultivation that may effectively help catfish farmers improve production under critical conditions caused by low temperatures or poor water quality in winter.

## 2. Materials and Methods

### 2.1. Cultivation and Preparation of the Probiotic Bacterium

A potent probiotic of the bacterial strain *Acinetobacter* KU011TH was obtained from the Laboratory of Aquatic Animal Health Management, Department of Aquaculture, Faculty of Fisheries, Kasetsart University, Bangkok, Thailand. The protocols for bacterial cultivation were based on descriptions by Bunnoy et al. [9,30].

### 2.2. Experimental Feed Preparation

The commercial feed Hi-Grade 9006 (Charoen Pokphand Food PCL., Bangkok, Thailand) is a type of floating pellet that was used as a basal diet for growing catfish during the 1st month of the growing period. The specific chemical compositions of the diets were analyzed, and the major composition of the diet was 42% crude protein, 10% moisture, 5% crude fat, and 3% crude fiber (Charoen Pokphand Food PCL., Thailand). In the 2nd–4th months, CP 9921 catfish feed (Charoen Pokphand Food PCL., Thailand) (25% crude protein, 12% moisture, 3% crude fat, and 8% crude fiber) was used. The amount of feed used was 2–10% of the fish body weight, which was adjusted weekly depending on the size of the fish. Five experimental diets were designed, including the control (A), which was top-dressed with 200 mL of 0.85% NaCl/kg feed (commercial feed without pre- and synbiotics). Diet B (prebiotic diet) was similarly prepared to the control diet (200 mL of 0.85% NaCl) and coated with 20 mL/kg feed chitosan (18% *w*/*w*) (Greater Vet Thailand, Nonthaburi, Thailand). The C, D, and E diets were mixed with 1 × 10^8^, 1 × 10^9^, and 1 × 10^10^ CFU of *Acinetobacter* KU011TH in 200 mL of 0.85% NaCl/kg diet and finally coated with chitosan at 20 mL/kg diet. No significant differences in nutritional profile among the tested diets were observed.

Routine preparation of the control, prebiotic, and synbiotic diets was conducted daily, and the diet was air dried at room temperature for approximately 1 h to remove the excess moisture. Then, chitosan was used to coat the dried feeds, which were allowed to air dry for approximately 30 min and were stored at 4 °C for all subsequent experiments.

### 2.3. Experimental Design, Animal Husbandry, and Ethics

A hybrid *Clarias* fingerling with an average body weight of 5.70 ± 0.41 g and initial total length of 8.80 ± 0.88 cm was obtained from Suphanburi Province (14°31′38.0″ N, 100°08′31.8″ E) and kept in a wet laboratory of the Center of Excellence in Aquatic Animal Health Management (CEAAHM), Department of Aquaculture Faculty of Fisheries, Kasetsart University, Chatuchak, Bangkok, Thailand. Fingerlings were acclimatized in a quarantine tank with fully aerated freshwater at a temperature of 28.0 ± 2 °C and pH of 7.5 ± 1 for 15 days before the experimental trial, while their health conditions were closely monitored as described in [9,30]. After the acclimatization period, 30 fish were dipped into a 100 ppm formalin solution for one minute and then randomly transferred to 20 experimental 250 L plastic tanks containing 200 L of water to provide 4 replicates of 5 treatments under completely randomized design (CRD) conditions. All the freshwater fish culture tanks were provided continuously with gentle aeration throughout the culture period. Water exchange was routinely performed every two days at 10–20% in all tanks. Tap water was stored in the reservoir to eliminate chlorine gas residues through aeration. Fish in each group were fed daily with 5 different experimental feeds, prepared as described above, twice a day (9:00 am and 16:00 pm) at 2–10% body weight depending on fish size. This indoor experiment was a 16-week trial conducted during the winter period from November 2021 to January 2022. The water temperature of the experimental tanks was not controlled, and the air temperature was changed during winter.

The experiment was conducted in accordance with the Ethical Principles and Guidelines for the Use of Animals National Research Council of Thailand and approved by the Animal Ethics Committee, Kasetsart University, Thailand (ACKU61-FIS-004).

### 2.4. Water Quality Analysis and Assessment

The water in the culture tanks and 3 m^3^ reservoir containers was collected weekly in polyethylene bottles at 24 h before water exchange and then maintained in a refrigerator at 4 °C until analysis. The temperature was recorded twice a day, in the morning (9:00 am) and in the evening (16:00 pm). At the time of sampling, the pH and dissolved oxygen (DO) were monitored using a portable Multimeter Instrument, SensoDirect 150 probe (Lovibond, Dortmund, Germany) immediately after water collection. Some essential chemical parameters, total ammonia nitrogen (NH_4_^+^ + NH_3_-N), nitrite nitrogen (NO_2_-N), and nitrate nitrogen (NO_3_-N), were measured using colorimetric and spectrophotometric methods, while total alkalinity and total hardness were quantified using the colorimetric titration methods of Baird et al. [38].

### 2.5. Fish Growth Parameters

There were no fish sacrificed during the study period, allowing the investigation of the impact of winter environmental factors on the growth of hybrid catfish. Data regarding fish growth were collected. Weekly, all remaining fish in each tank of each treatment were weighed by an electrical balance, and the total length was also recorded. Fish growth parameters were calculated, including total weight gain (WG) or length gain (TLG), average daily growth (ADG), relative growth rate (RGR), and specific growth rate (SGR). Feed and mortality in each treatment were recorded daily to calculate the feed conversion ratio (FCR) and cumulative mortality, respectively. The formulas used were as follows:
WG (g) = Wt − WiTLG (cm) = TLt − TLiADG (g/individual/day) = (Wt − Wi)/tSGR (%/day) = log (Wt) − log (Wi)/t ×100FCR = amount of total feed given/WGAccumulated mortality (%) = 100 × (number of dead fish)/(total initial number of fish)
where Wi is the initial weight, Wt is the weight at the target sapling time during the trial, TLG is the final total length, TLi is the initial total length, and TLt is the total length of the sapling at the collection time during the trial.

### 2.6. Health Status, Serum Immune Parameters, and Immune-Related Gene Expression Analysis

At the end of the 16-week feeding trial, eight fish (2 fish/tank) from each treatment were collected. Blood samples from selected fish were withdrawn for the analysis of serum immune parameters, hematological parameters, and immune-related gene expression. The intestine and liver were dissected for histopathological analysis. Whole blood, head kidney, spleen, and liver tissues were used to study immune-related gene expression. DNA from the intestine was isolated for quantitative evaluation of the copy number of the target probiotic strain’s genes. Details of each analysis are described below.

#### 2.6.1. Serum Immune Parameters

##### Blood Collection

Fish from each treatment group were randomly sampled for whole blood withdrawal from the caudal vein using a syringe with a 23 G needle without anticoagulant. Five microliters and fifteen microliters were used for total blood count/differentiated count and hematocrit, which are described below. The remaining sample blood was allowed to clot in an inclined position for 2 h at room temperature. Then, centrifugation was performed at 8500 rpm for 10 min at 4 °C. The centrifuged whole blood was preserved with 1 mL of TRIzol reagent (Thermo Fisher Scientific, Waltham, MA, USA) at −80 °C for total RNA extraction for immune-related gene expression analysis. The serum was stored and frozen at −20 °C until immunological analysis.

##### Respiratory Burst Analysis

The respiratory burst activity of phagocytes was measured using nitro-blue tetrazolium (NBT) dye reduction to quantitate superoxide anion, which is one of the initial reactive oxygen intermediates (ROIs). Duplicates of each sample, 100 μL of fish serum, were loaded into a 96-well plate. The substrate, NBT solvent (1 mg/mL normal saline), was added (14.29 μL) into each well and then incubated at 37 °C for 2 h. Finally, the absorbance of the solution was read in a microplate reader (iMark™ Microplate Absorbance Reader, BIO-RAD, Tokyo, Japan) at a wavelength of 655 nm against distilled water as a blank [39].

##### Lysozyme Activity Assay

Lysozyme activity was measured using turbidimetric assays. A total of 10 μL of fish serum was mixed with 250 μL of *Micrococcus lysodeikticus* (Sigma-Aldrich, Darmstadt, Germany) as a substrate at a stock concentration of 0.2 mg/mL in PBS (pH 6.2) to obtain a final volume of 260 μL per well. The absorbance was recorded at a wavelength of 450 nm in different periods (0 min (without *M. lysodeikticus*) and 5 min after adding *M. lysodeikticus*) at room temperature using an iMark™ Microplate Absorbance Reader (BIO-RAD). One unit of enzyme activity was defined as the volume of the enzyme that catalyzed a reduction in absorbance of 10^−3^/min [40].

##### Hemolytic Activity by the Alternative Complement Pathway (ACH_50_) Assay

Hemolytic activity was determined using the alternative complement pathway (ACH_50_) assay; 200 μL of serum was added to the wells of a 96-well plate (curved-bottom) in the second column. Next, 100 μL of phosphate-buffered saline (PBS, pH 7.4) was added to the remaining columns for serum reaction. Then, 100 μL of serum was diluted with PBS in wells, and two serial dilutions were carefully added to the last well. Consequently, 100 μL of sheep red blood cells (ShRBCs, stock concentration 1 × 10^8^ cells/mL) as target cells for hemolysis were added to each diluted serum well. The serially diluted plasma of each serum sample was incubated with ShRBCs for 90 min at 32 °C and centrifuged at 1500× *g* for 5 min at RT to collect the supernatant. Then, 100 μL of the supernatant was transferred to new flat-bottomed 96-well plates, and the absorbance of the supernatant was measured at 540 nm. Incubation of ShRBCs with distilled water was performed as a positive control (100% lysis), and ShRBCs without serum were used as blanks or negative controls (0% lysis). The reciprocal of the serum dilution that caused 50% lysis of ShRBCs was designated ACH_50_ [41].

##### Bactericidal Activity Analysis

A single colony of *A. hydrophila* (AQAH001) from the original stock, which was isolated from infected hybrid catfish and stored in CEAAHM, was subcultured in 50 mL of nutrient broth (NB) at 35 °C and 160 rpm for 18 h. The cell pellets were harvested by centrifugation at 2500 rpm for 10 min. A bacterial suspension with a concentration of 1 × 10^4^ CFU/mL in 0.85% NaCl was prepared by dilution of the cell suspension with an absorbance of 1.0 (concentration of 5 × 10^8^ CFU/mL, OD 600 nm). A total of 100 μL of fish serum was mixed with 100 μL of a bacterial suspension of *A. hydrophila* (prepared above) in 96-well plates and incubated at 37 °C for 2 h prior to evaluation of survival by inoculating 100 μL of the mixture on plate count agar (PCA) for 18 h. Samples without bacteria and serum served as the blanks (negative control) and positive controls (100% survival or 0% bactericidal activity), respectively. The bactericidal activity of the experimental fish serum was assessed by determining the survival of bacteria after incubation with serum [42].

#### 2.6.2. Hematological Analysis

##### Hematocrit

Hematocrit or erythrocyte pack cell volume was determined in whole blood. Fifteen microliters of freshly withdrawn blood were loaded into a microcapillary tube with anticoagulant to reach eight-tenths of the tube length. Then, plasticine was placed at the bottom of the tube at one-tenth of the tube length and centrifuged using a microhematocrit centrifugation machine (Suranaree Medical Equipment Co., Ltd., Nakhon Ratchasima, Thailand) at 12,000 rpm for 2 min. The percent hematocrit was read on a standard hematocrit chart (Suranaree Medical Equipment Co., Ltd., Nakhon Ratchasima, Thailand) and recorded.

##### Total Blood Count and Differentiated Count

Blood cell staining was performed immediately after sampling. In a vial, a 5 μL blood sample was eluted with 995 µL of Natt–Herricks staining solution (dilution ratio, 1:200) for at least 5 min. Fifteen microliters of mixed solution were loaded on a hemocytometer with 2 replicates. The total blood count and differentiated blood count containing red blood cells (RBCs) and white blood cells (WBCs) were implemented according to the procedure of a previous study [43].

#### 2.6.3. Histopathological Analyses

Tissue specimens from the middle intestine and liver of fish were collected and fixed in 10% neutral-buffered formalin for the first 24 h, and a new buffer formalin exchange was performed. The fixed tissues were processed using automatic tissue processing, involving dehydration, clearing, and embedding in paraffin wax. Then, the embedded tissues were cut into 4–5 μm sections using a YD-315 rotary microtome (Jinhua YIDI Medical Appliance Co., Ltd., Jinhua, China), stained with standard hematoxylin and eosin, and mounted on clean glass slides. Histological alterations were examined using light microscopy. Villus and muscularis length and the density of goblet cells were also evaluated (Nikon, Eclipse, E 800, Tokyo, Japan) [21,30].

#### 2.6.4. Effects of Synbiotics on the Expression of an Immune-Related Gene Using Quantitative Real-Time–PCR (qRT–PCR) 

##### Total RNA Extraction of the Target Tissues

The same sample fish was sacrificed to obtain the head kidney, liver, spleen, and whole blood samples. Immediately, the four tissue samples (approximately 100 mg) were preserved with 1 mL of TRIzol reagent (Thermo Fisher Scientific, MA, USA) at −80 °C. Total RNA was extracted using an automatic tissue extractor (MP, Irvine, CA, USA) following the manufacturer’s instructions. The obtained pellet of total RNA was suspended in 50 µL of RNase-free water, and the total RNA concentration was quantified by reading the absorbance at wavelengths of 260 and 280 nm using a Nanodrop spectrophotometer (Thermo Fisher Scientific, MA, USA).

##### First-Strand cDNA Synthesis

Total RNA samples (1 μg/μL) were used for cDNA synthesis using a high-capacity cDNA reverse transcription kit (ReverTra Ace^®^ qPCR RT Master Kit with gDNA Remover) according to the manufacturer’s instructions (Toyobo Co., Ltd., Osaka, Japan). Briefly, RNA templates were incubated for 5 min at 65 °C. DN Master Mix (4×) was gently mixed with gDNA Remover at a ratio of 440 μL:8.8 μL and incubated for 5 min at 37 °C. Afterward, the reagents were added to 5×RT Master Mix II and incubated for 15 min at 35 °C and then for 5 min at 50 °C. Finally, the reaction was terminated by heating at 98 °C for 5 min, and the cDNAs were stored at −20 °C until further analysis.

##### Quantitative Real-Time PCR Analysis (qRT–PCR)

One microliter of first-strand cDNA from each tissue of treated and control fish was subjected to qRT–PCR analysis using an Mx3005P real-time PCR system (Stratagene, La Jolla, CA, USA) with Systematic software (version 4.0) and Brilliant II SYBR Green qPCR Master Mix (Stratagene, USA) according to the manufacturer’s protocol. Specific primer pairs for the alpha-2-macroglobulin (*Cm*-A2M), bactericidal permeability-increasing protein (*Cm*-BPIP), CC chemokine (*Cm*-CC), C3 chemokine (*Cm*-C3), lysozyme C (*Cm*-LYZC), myeloperoxidase (*Cm*-MYE), and nuclear factor kappa B (*Cm*-NF-κB) genes were used, and all qPCR conditions strictly followed the study of Bunnoy et al. [9]. qRT–PCR to measure β-actin gene expression was conducted and used for normalization of the expression levels of immune-related genes. The threshold cycle value differences (ΔCt) between the immune-related and *β-actin* genes in each organ were calculated using the ΔΔCt method [9,44]. The relative expression ratio of each target gene was calculated according to the 2^–ΔΔCt^ method [44].

### 2.7. qPCR Analysis for Determination of the Probiotic Copy Numbers in the Intestine of the Sample Catfish

Intestinal genomic DNA (100 μg) from 4 catfish in each treatment was extracted using DNAzol^®^ (Thermo Fisher Scientific, USA) according to the manufacturer’s instructions. The genomic DNA, 1 µL (100 ng/μL), was used for qPCR analysis using *Acinetobacter* KU011TH *Acgyr*B primers for absolute quantification by following the methods described previously [30].

### 2.8. Effects of Prebiotics and Synbiotics on Disease Resistance against Aeromonas Hydrophila

#### 2.8.1. Experimental Fish

Ten fish from each replicate tank of each treatment were transferred to 20 new 250 L plastic tanks. Fish of each treatment were acclimatized in the same conditions as those used in a previous experiment for 7 days. During the experimental period, fish were fed their experimental feed.

#### 2.8.2. Challenge Test

At day 8 after acclimatization, every fish in each tank was intraperitoneally injected with 0.1 mL of bacterial solution containing 1 × 10^8^ CFU/mL *A. hydrophila* (AQAH001), which was previously prepared as described above. All injected fish were transferred back to their experimental tanks and husbandries under the same conditions as those used during the acclimatization period. The behaviors and mortality were constantly recorded every 24 h for 14 days. Furthermore, the liver, head kidney, and spleen tissues of abnormal fish were collected and analyzed for histological alterations following a previously described method [30].

### 2.9. Data and Statistical Analysis

The water parameters, growth parameters, innate-immune indexes, and expression of immune-related genes were statistically analyzed by one-way analysis of variance (ANOVA) to determine significant differences among the tested groups, and polynomial contrasts were further analyzed to determine the effects of *Acinetobacter* application at different doses. Student’s *t*-test was employed to indicate significant differences between the control and treatments using the Statistical Package for Social Science (IBM SPSS statistic version 26). All data were plotted using Microsoft Excel (MS Excel 2016), and the results are presented as the mean ± SD. Significant differences were considered when *p* < 0.05. Survival data from the challenge test against *A. hydrophila* (AQAH001) were used for Kaplan–Meier survival analysis using SPSS for Mac (version 24.0, Chicago, IL, USA). The level of statistical significance of survival rates between the control and treatment groups was indicated using Student’s *t*-test at the significance level of 0.05.

## 3. Results

### 3.1. Water Quality Analysis

On average, water parameters, such as dissolved oxygen (5.04 ± 0.12 mg/L), total hardness (237.34 ± 11.94 mg/L as CaCO_3_), pH (7.35 ± 0.36), total ammonia-N (1.19 ± 0.03 mg NH_3_-N/L), nitrite-N (1.24 ± 0.06 mg NO_2_-N/L), and nitrate-N (3.44 ± 0.29 mg NO_3_-N/L) levels, varied during the trials. However, these ranges were used as optimal conditions, despite significant differences observed among treatments in certain weeks (*p* < 0.05). No severely critical effects of these water parameters on catfish were observed during the testing periods (Appendix A). Two key water parameters, temperature (27.47 ± 0.55 °C) and total alkalinity (28.43 ± 0.83 mg/L as CaCO_3_), dramatically decreased during weeks 8 and 2, respectively (Appendix A and Figure 1a). To maintain minimal levels of alkalinity (>20 mg/L as CaCO_3_), CaCO_3_ was continuously supplied to the reservoir of all experimental units until the end of the experiment. A decrease in temperature close to 25.0 °C at the beginning of week 8 strongly affected the physiological responses of fish by stabilizing the feeding rate, as indicated by the feed intake (Figure 1b).

### 3.2. Growth Performance Analysis and Mortality

After 16 weeks of application, most growth parameters of groups B and C were observed to be higher than those of the other groups. However, it was found that fish in groups B, C, and D exhibited a significantly higher FW and SGR than group E (*p* < 0.05) but not the control. The FW of groups B, C, and D was 164.97 ± 11.28, 167.51 ± 22.82, and 156.36 ± 10.93, respectively, while that of groups A and E was 157.51 ± 13.80 and 131.44 ± 3.66, respectively. The SGRs of groups B, C, and D were 1.28 ± 0.26, 1.25 ± 0.27, and 1.24 ± 0.24, respectively, while those of groups A and E were 1.24 ± 0.19 and 1.15 ± 0.31, respectively (Table 1). These two parameters showed a quadratic trend when polynomial contrast analysis was conducted (*p* < 0.05).

### 3.3. Hematological Analysis

At the end of the trials, the packed cell volume of fish in all groups was in the normal range. The value for group D was the highest at 38.33 ± 4.51% and significantly higher than that for fish in group C, with the lowest value of 30.10 ± 1.00% (*p* < 0.05) (Figure 2a). Regarding the complete blood count, no significant difference in red blood cells among the tested groups was observed, while the white blood cell counts of groups E and D were significantly higher than those of the other groups (A and B) but not that of group C (Figure 2b). Similar results were also observed for the differentiated blood count, in which the thrombocyte and monocyte counts of groups E and D were significantly higher than those of groups A and B. Interestingly, the lymphocyte counts of all treatments with chitosan with added probiotics were significantly higher than those of the control and prebiotic treatment groups (*p* < 0.05) (Figure 2c). Additionally, all significant parameters showed a linear contrasting trend, wherein the trends for WBCs and thrombocytes were significant (*p* < 0.05), while those for monocytes and lymphocytes were highly significant (*p* < 0.01).

### 3.4. Serum Immunological Analysis

The NBT dye reduction of the synbiotic-treated D group was significantly higher than that of the A (control), B, and E groups but not different from that of group C (*p* < 0.05) (Figure 3a). The lysozyme activity of symbiotic group D was significantly higher than that of the other groups (*p* < 0.05), and that of synbiotic group C was significantly higher than that of the control group (A) and prebiotic group B (*p* < 0.05); however, it was not higher for synbiotic group D (*p* > 0.05) (Figure 3b). Interestingly, the ACH_50_ of the synbiotic C (83.70 ± 8.70) and D (84.46 ± 9.45) groups was highly significantly higher than those of the other groups (A, B, and E; 57.79 ± 13.74, 47.37 ± 10.47, and 59.39 ± 5.86, respectively) (*p* < 0.01) (Figure 3c). Additionally, the serum bactericidal activities of a prebiotic (B) group and all tested synbiotic (C, D, and E) groups exhibited significant relative bactericidal activity against a pathogenic bacterium (*A. hydrophila*) compared to that of the control group (*p* < 0.01) (Figure 3d). Furthermore, all four serum immunological parameters were observed as cubic contrast trends, wherein NBT and bactericidal activity showed significant trends (*p* < 0.05), while lysozyme and ACH_50_ activities showed highly significant trends (*p* < 0.05).

### 3.5. Histological Analysis

Intestinal histological structure analysis revealed that the fish in the prebiotic group (B) and the synbiotic-treated groups (C and D) displayed better villus height and intestinal wall thickness, respectively (Figure 4). Fish in prebiotic group B and synbiotic groups C and D had significantly higher villus heights than those in control groups A and E (*p* < 0.05), with values of 0.31 ± 0.06 mm, 0.28 ± 0.02 mm, 0.30 ± 0.08 mm, 0.11 ± 0.05 mm, and 0.11 ± 0.03 mm, respectively (Figure 5a). Similarly, the intestinal wall thickness of each group showed the same trend of 0.07 ± 0.02 mm, 0.06 ± 0.02 mm, 0.07 ± 0.01 mm, 0.03 ± 0.02 mm, and 0.03 ± 0.01, respectively. However, the goblet cell density was the highest in synbiotic group C (86.25 ± 17.95 cells/500 μm^2^), which was significantly higher than those of the control and other groups, which ranged between 46.50 ± 3.42 and 58.50 ± 12.61 cells/mm^2^ (Figure 5b). For the livers, all tested groups showed typical characteristics of normal histology indicating normal health status (Figure 6).

When polynomial contrasts were analyzed, villus length and goblet cells belonged to cubic forms (*p* < 0.05 and *p* < 0.001, respectively), while the muscularis layer showed a quadratic contrasting trend (*p* < 0.001).

### 3.6. Quantification of the Copy Number of the Gyrase Subunit B Gene of Acinetobacter KU011TH in the Intestine of Experimental Catfish

The DNA copy numbers of the gyrase subunit B gene of *Acinetobacter* KU011TH were analyzed using absolute qPCR analysis. Unexpectedly, it was found that the absolute bacterial DNA quantity in the midgut of synbiotic group C fish had the highest values of 6762.97 ± 1168.17 copies/100 ng, which was significantly higher than that in the other groups, followed by group E with 388.07 ± 243.82 copies/100 ng (*p* < 0.001). The other groups only showed very low copy numbers, ranging between 23.85 ± 31.62 and 200.08 ± 277.29 copies/100 ng (Figure 7). The trend of the gyrase subunit B gene copy number strongly decreased in a quadratic manner (*p* < 0.001).

### 3.7. Immune-Related Gene Expression Analysis

The fold changes in the relative expression of all tested immune-related genes in whole blood are shown in Figure 8a–g. The *BPIP*, *C3*, *NF-κB*, and *LYZC* genes were significantly upregulated, especially in the two synbiotic groups C and D. In particular, the *NF-κB* gene exhibited high expression levels of 65.60 ± 40.19 and a 51.70 ± 4.05-fold change compared with the control and other groups, which exhibited low expression levels of 0.80 ± 0.32 to 3.03 ± 4.72 (*p* < 0.05). In the head kidney (Figure 9a–g), synbiotic groups D and E showed strong induction of the expression of all the tested genes except the *NF-κB* gene. Among the tested genes, the *MPO* gene showed very high expression in groups D and E, with the expression increasing 61.95 ± 37.38- and 96.36 ± 62.28-fold, respectively, compared with that in the control and other groups, which had low expression levels of 1.89± 1.75 to 27.60 ± 39.35 (*p* < 0.05) (Figure 10d). In the spleen, synbiotic group D showed strong induction of almost all the target genes, especially the *α2M*, *CC*, *BPIP*, *MPO*, *C3*, and *NF-κB* genes (except the *LYZC* gene), with the highest expression levels of 11.11 ± 2.61-, 67.05 ± 27.42-, 29.50 ± 17.17-, 84.69 ± 46.24-, 50.36 ± 46.67-, and 70.28 ± 33.51-fold compared with the control, respectively (*p* < 0.05) (Figure 10a–f). Finally, all the tested genes were found to be expressed at very low levels in the liver compared with other organs. Significant differences were found in the expression of the *NF-κB* and *LYZC* genes, for which only fish in group E showed significant differences, at 9.30 ± 6.03- and 12.18 ± 5.60-fold compared with the control, respectively (*p* < 0.05) (Figure 11f,g).

In whole blood cells, the expression of the *BPIP* and *LYZC* genes showed cubic contrast (*p* < 0.05), while *C3* and *NF-κB* showed quadratic trends (*p* < 0.01). In the head kidney, *α2M*, *CC*, *MPO*, *C3*, and *LYZC* were evidently expressed with linear contrasts (*p* < 0.01), and BPIP and NF-κB showed cubic trends (*p* < 0.05). In the spleen, *CC*, *NF-κB*, and *LYZC* were expressed in linear forms (*p* < 0.01), but BPIP showed a cubic trend (*p* < 0.05). Finally, in the liver, only *α2M* and *LYZC* were expressed with cubic (*p* < 0.01) and linear contrast (*p* < 0.05) trends, respectively.

### 3.8. Disease Resistance against A. hydrophila

Survival analysis clearly indicated that death occurred in all groups on the first day after injection and began to decrease significantly after the second day. On the eighth day, the death of the experimental fish was initially stopped and remained stable until the end of the experiment (Figure 12). On day fourteen, the synbiotic group E exhibited a significantly (*p* < 0.05) lower survival rate (at 55.33 ± 8.82%) than group A (control group) and group D, with survival rates of 74.00 ± 6.50% and 72.83 ± 5.70%, respectively. Groups B and C showed the highest survival rates of 84.33 ± 2.21 and 79.50 ± 6.34%, respectively, which were significantly higher than those of the other groups (*p* < 0.05). All abnormal fish obviously developed ascites, dark coloration, fin rot, and skin hemorrhage, and many yellowish colonies developed, as determined by clinical diagnosis under laboratory investigation.

## 4. Discussion

The current study contributes to the growing research on probiotics, prebiotics, and synbiotics for eco-friendly aquaculture. Specifically, it focused on exploring the potential applications of combining *Acinetobacter* KU011TH probiotics with chitosan, which has strong prebiotic properties and coating efficiency [45,46], as a synbiotic for use in catfish aquaculture [47,48]. This study is believed to be the first to investigate the supplementation of the genus Acinetobacter with chitosan and its use as a potential synbiotic in aquatic vertebrates under low-temperature and low-alkalinity conditions. These are also novel applications in *Clarias* and *Clarias* hybrid catfish, for which probiotics, prebiotics, and synbiotics were found to be seldom used in aquaculture [20,21]. The use of synbiotics in aquaculture has had positive effects on growth performance, immunity, disease resistance, and water quality improvement [12,32,49]. These factors greatly impact survival, growth rates, and overall production. However, aquaculture systems face natural challenges during certain periods [50,51], particularly in Thailand and other Southeast Asian countries, where low temperatures and heavy rains during early winter can disrupt water quality parameters such as alkalinity [52,53]. These factors contribute to production reductions due to growth retardation and severe bacterial diseases during this period.

In the current study, the effects of various concentrations of *Acinetobacter* KU011TH and a fixed concentration of chitosan were investigated via long-term application. Based on polynomial contrast analysis, it was shown that the effect of *Acinetobacter* KU011TH application mostly reflected linear and cubic trends, indicating both positive and negative effects of higher dose applications that evidently impacted growth and physiological and immune responses in different manners.

Aquaculture systems expose fish to various environmental stressors, triggering diverse biological responses [51,54]. As poikilothermic animals, fish rely on external temperatures, and lower water temperatures typically reduce their appetite and nutrient absorption rate [55,56]. Temperature also affects important processes such as gut juice secretory operation (via food ingestion), GIT motility, digestive enzyme activity, and digestion and absorption rates [57,58]. In this context, the current study was carried out at a suboptimal water temperature (25 ± 0.5 °C) during winter. Basically, *Clarias* catfish can consume food and grow well at temperatures ranging from 26 to 32 °C [59,60], and the reduced dietary intake may result in suppressed physical and biological responses caused by inflammation in various body tissues. Our findings revealed that the water temperature rapidly dropped below 25 °C during November and December and stabilized at 25–26 °C, directly impacting the feeding rate of the experimental fish. In the early period of week 2, the alkalinity of the tap water decreased to a very low level (less than 20 mg/L CaCO_3_), caused by natural water sources being affected by heavy rain during the monsoon season. The application of CaCO_3_ effectively elevated the alkalinity in the reservoir before the water was supplied to all the experimental tanks. Basically, it is crucial to maintain alkalinity above 20 mg/L CaCO_3_, as it plays a vital role in maintaining adequate buffering capacity [61].

The application of chitosan and chitosan plus *Acinetobacter* KU011TH probiotics at different levels to create effective synbiotics indicated that these treatments could not improve most growth parameters of hybrid catfish during winter when the temperature and alkalinity were low, except for the final length and specific growth rate, for which chitosan group B and synbiotic groups C and D exhibited significantly higher values than the control group and group E at the highest probiotic application dose. These obtained data were supported by histological changes in the intestines, which may indicate the relationship between prebiotic or synbiotic application and significant growth rates.

Active intestines exhibit physiological changes reflected in improvement in biological markers such as increased muscular layer thickness, intestinal villus height, and goblet cell densities. These improvements in intestinal morphometries, including the muscularis layer and villi height, contribute to enhanced digestion, absorption, and immune defense mechanisms, ultimately promoting growth performance [62,63,64]. Goblet cells (GCs), specialized epithelial cells found on mucosal surfaces, play a crucial role in maintaining intestinal barriers and secreting antimicrobial proteins, chemokines, and cytokines that contribute to mucosal immunity [65,66]. Higher levels of GCs on the mucosal membrane are associated with improved nutritional absorption and immunity in fish [67]. In this study, an increased number of GCs in the midgut indicates higher immunity in fish fed a synbiotic diet than in fish fed a control diet.

The beneficial properties and physiological functions of chitosan and the added *Acinetobacter* KU011TH have been observed to promote the growth of healthy bacteria in the host gut [9,15,68]. The encapsulation property of chitosan aids in the gastrointestinal digestion and adherence of *Acinetobacter* KU011TH in the distal gut, thereby influencing the morphology of the intestinal organs and increasing the nutrient absorption area. This finding is consistent with a study by Salam et al. [36], which demonstrated that chitosan improved the gut microbiota and internal organ morphology, leading to enhanced growth in juvenile Barbonymus gonionotus. Additionally, the positive impacts of nutrients could benefit the host via gut microbiota composition and modulate GI development [12].

The application of the chitosan and synbiotic was further investigated for effects on the health status of the tested hybrid catfish, as indicated by hematological indexes, histopathology of the liver and intestine, serum immune parameters, and expression analysis of seven immune-related genes. Application of the synbiotic in groups D and E resulted in a significant enhancement of immune parameters compared with other groups, except in liver histology. When all experimental groups were challenged with *A. hydrophila*, fish in groups B and C showed significantly higher survival rates than the control group and synbiotic D group, while the survival of group E was significantly lower than that of the other groups. This suggests that the application of synbiotics at high doses of 1 × 10^9^–1 × 10^10^ CFU/kg diet may induce some abnormalities in fish.

Based on the above information, it seems that the long-term application of synbiotics with high doses of *Acinetobacter* KU011TH in groups D and E may have generated improper conditions in the fish. This phenomenon is strongly supported by studies performed on other fish species, including Nile tilapia, which was infected with Streptococcus agalactiae by Wu et al. [69]. Transcriptome analyses also revealed stronger adaptive immunity in the gut and stronger innate immunity in the liver. Significant upregulation of immune-related genes in the gut and liver of healthy (homeostasis) and severely infected (inflammation) fish was clearly observed. The very upregulation of gene groups associated with acute phase protein, pattern recognition, complement system, inflammatory cytokines, and T/B-cell antigen activation, especially α2M, C-type lectin, C3 complement, and CC chemokines, was similar to the results of our study. Additionally, the inflammatory conditions of fish in groups D and E were evidently supported by the significantly high levels of white blood cells, particularly thrombocytes, monocytes, and lymphocytes, in the hematological study. This condition can consequently affect disease resistance, making the fish more vulnerable to invading pathogens [69], as indicated by the significantly lower survival rate after the challenge with *A. hydrophila* in the challenge test. This suggests that the long-term application of synbiotics with high doses of *Acinetobacter* KU011TH probiotics must be considered for further use.

Fish employ innate and adaptive immune mechanisms involving humoral and cellular responses to combat microbial invasion [70]. Nonspecific humoral factors, such as growth-inhibiting substances, antimicrobial peptides, and complement peptides, and nonspecific cellular responses, such as phagocytosis, initiate immune defenses. In specific disease responses, specialized lymphoid cells and immunoglobulins play crucial roles in adaptive mechanisms [70,71]. Therefore, some key components applied in the tested diet may have exerted their beneficial functions in the current study. The prebiotic-like chitosan enhances protease, lipase, and amylase activities in the gut, which are important for digestion and contribute to a healthy gut microbiota [72]. Gram-negative bacterial glycoconjugates, including peptidoglycan (PGN) and lipopolysaccharide (LPS), may serve as pathogen-associated molecular patterns (PAMPs) on the cell wall. *Acinetobacter* KU011TH contains abundant LPS to stimulate both humoral and cellular defenses in the host [9,30]. PAMPs or microbe-associated molecular patterns (MAMPs) interact with pattern recognition receptors (PRRs) that activate signaling pathways to enhance the host immune system. Additionally, various compounds produced by probiotics, such as bacteriocins, H_2_O_2_, NO, organic acids, indole, and secreted proteins, strengthen the gut epithelial barrier by promoting mucus secretion, increasing antimicrobial peptide production, and enhancing tight junction protein expression [65,66,73]. The intestinal epithelial cells (IECs) that form the monolayer of the gut create a barrier between the host internal organs and the microbiota. Disruption of this barrier, often caused by dysbacteriosis, can lead to intestinal disorders, including inflammatory bowel disease and necrotizing enterocolitis [57]. While gut diseases are influenced by diet, genetics, and the environment, dysbacteriosis is widely considered to impact the integrity of the intestinal barrier [65].

Therefore, an increasing survival rate after the *A. hydrophila* challenge implied that treatment with chitosan (group B) and synbiotics (group C) could greatly trigger catfish immunity via several innate and adaptive mechanisms, as demonstrated by increased immune-related gene expression and the ability to resist pathogenic bacteria after long-term, 16-week application during the winter. These results strongly support the prebiotic properties of chitosan, which is similar to the results of previous studies on Barbonymus gonionotus [36], hybrid grouper [74], and sea bass [75]. The combination of chitosan and a lower dose of *Acinetobacter* KU011TH (1 × 10^8^ CFU/kg diet) in the current study is showing better efficacy than *Acinetobacter* KU011TH application alone in terms of both health status and disease resistance against *A. hydrophila* [9]. This finding is novel and goes beyond the previous research by Bunnoy et al. [9] on the effects of probiotic *Acinetobacter* KU011TH (at 1 × 10^9^ CFU/kg diet) administration that improved catfish growth, health performance, innate humoral and cellular responses, immune-related gene expression, skin physical defense, and disease resistance under normal culture conditions.

## 5. Conclusions

In summary, the current study revealed that the combination of chitosan and *Acinetobacter* KU011TH (at concentrations of 1 × 10^8^ and 1 × 10^9^ CFU/kg feed to hybrid catfish feed) cannot contribute improved growth performance, and adverse effects on growth rates were found at the highest concentration of 1 × 10^10^ CFU/kg feed compared with the control group during winter conditions. However, long-term application of all symbiotic groups significantly increased the expression of immune-related genes and improved serum immune parameters. Finally, the application of only chitosan and synbiotic group C significantly improved disease resistance against *A. hydrophila*. Ultimately, the data obtained from all synbiotic treatments in this study are crucial for developing concepts that may profitably optimize fish production in aquatic industries by elevating growth and improving health, particularly immunological responses, and chitosan and its combination with Acinetobacter KU011TH at 1 × 10^8^ CFU/kg feed are recommended for optimizing growth and immune responses and stronger disease resistance against *A. hydrophila* infection in *Clarias* hybrid fish during winter. Notably, the results of in vitro experiments differed from those of the challenge test in vivo, suggesting the presence of unknown factors in the serum of chitosan- and synbiotic-treated groups that effectively eliminated the pathogenic bacterium. Therefore, further investigation is required to understand the correlations among immune responses, gene expression, and response in challenge tests. Further research in field trials should be conducted to examine suitable effective doses or times, cost-effectiveness, and the mechanisms underlying the effects on growth efficiency, health, metabolism, and disease resistance.

## Figures and Tables

**Figure 1 microorganisms-11-01807-f001:**
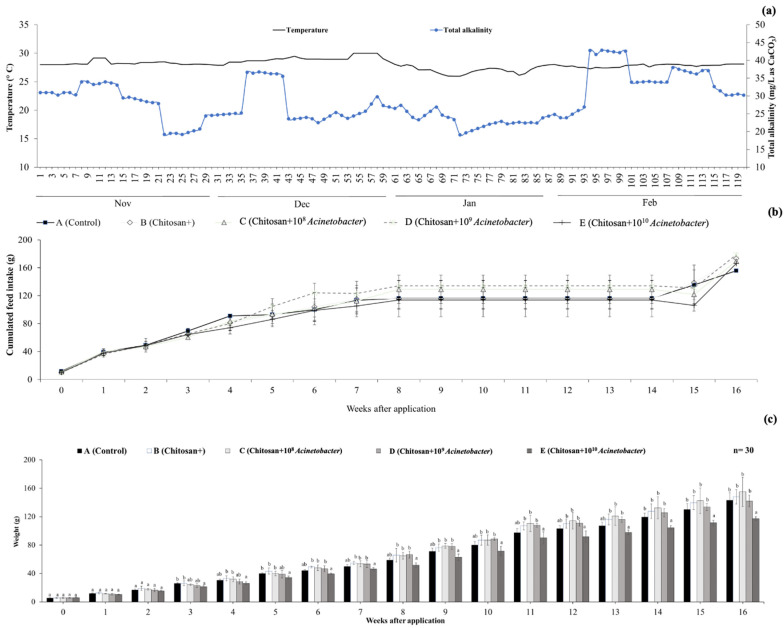
Average temperature and alkalinity of water during experimental periods (**a**). Total weight (**b**) and total length (**c**) of catfish fed different diets. Different letters on each bar indicate significant differences among the tested groups (*p* < 0.05).

**Figure 2 microorganisms-11-01807-f002:**
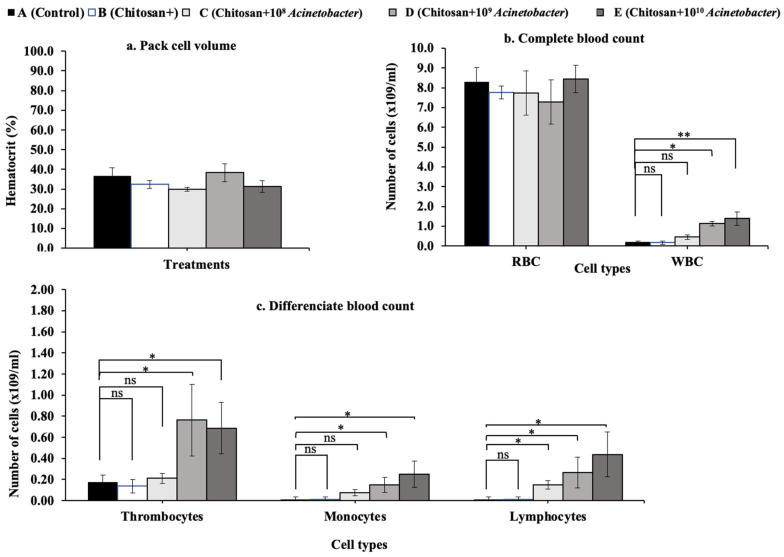
Hematological parameters; hematocrit (**a**), complete blood count (**b**), and differentiated count (**c**) of experimental hybrid catfish after feeding with experimental feed. Asterisks * and ** on each graph are considered significant at *p* < 0.05 and *p* < 0.01, respectively (*n* = 4), and ns indicates nonsignificant compared with control (*p* > 0.05).

**Figure 3 microorganisms-11-01807-f003:**
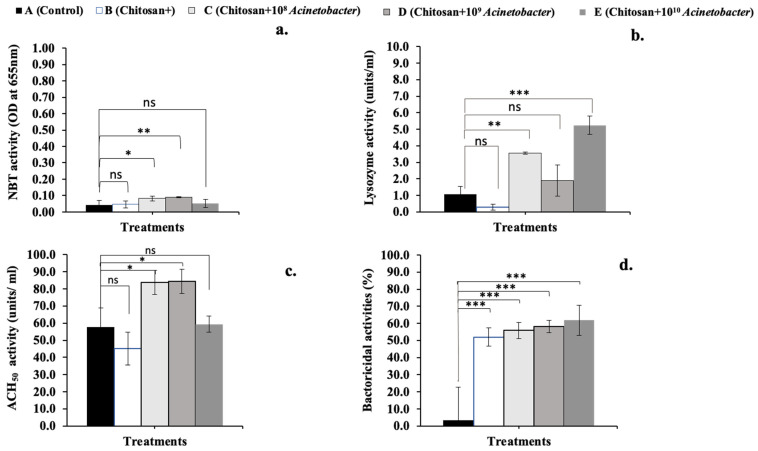
Effect of long-term synbiotic application on serum immune responses of hybrid catfish; NBT dye reduction (**a**), lysozyme activity (**b**), alternative complement pathway hemolytic activity (ACH_50_) (**c**), and bactericidal activity (**d**). Asterisks *, ** and *** on each graph are considered significant at *p* < 0.05, *p* < 0.01 and *p* < 0.001, respectively (*n* = 4), and ns indicates nonsignificant compared with control (*p* > 0.05).

**Figure 4 microorganisms-11-01807-f004:**
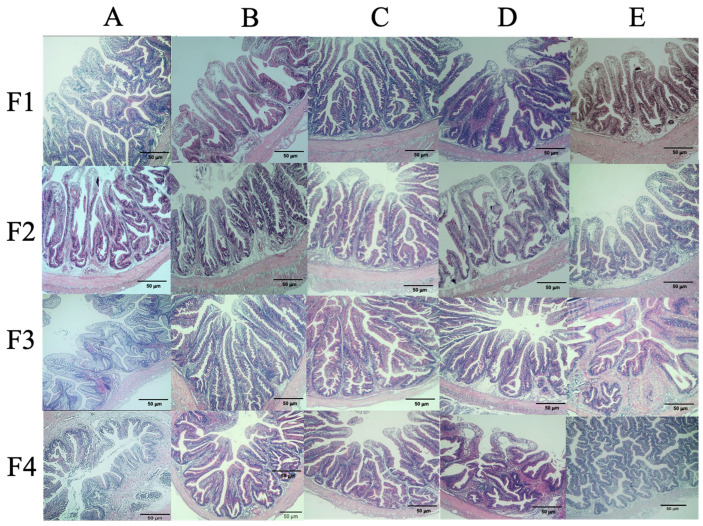
Histological analysis of the intestine of hybrid catfish fed control feed and supplemented diets with different experimental feeds. A–F indicate control, chitosan, chitosan+ 1 × 10^8^, chitosan+ 1 × 10^9^ and chitosan+ 1 ×10^10^ CFU/kg groups, respectively and F1–F4 indicate the number of fish in each sample.

**Figure 5 microorganisms-11-01807-f005:**
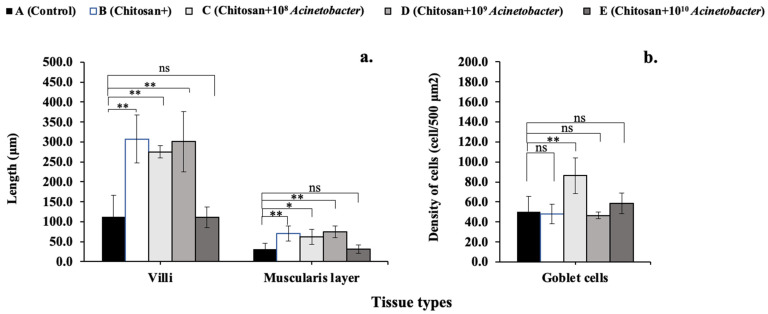
Histological analysis of villi and muscular layer length (**a**) and goblet cell density (**b**) of hybrid catfish fed control feed and supplemented diets with different experimental feeds. Asterisks * and ** on each graph are considered significant at *p* < 0.05 and *p* < 0.01, respectively (*n* = 4), and ns indicates nonsignificant compared with control (*p* > 0.05).

**Figure 6 microorganisms-11-01807-f006:**
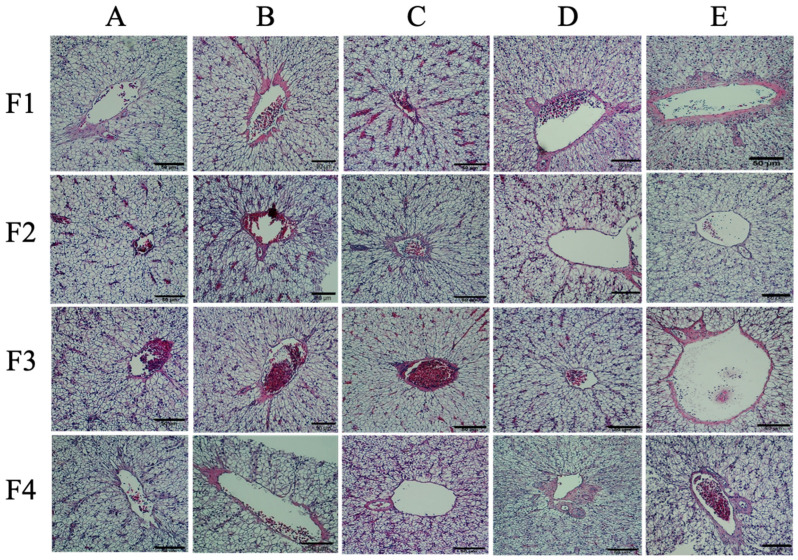
Histological analysis of the liver of hybrid catfish fed control feed and supplemented diets with different experimental feeds. A–F indicate control, chitosan, chitosan+ 1 × 10^8^, chitosan+ 1 × 10^9^ and chitosan+ 1 × 10^10^ CFU/kg groups, respectively and F1–F4 indicate the number of fish in each sample.

**Figure 7 microorganisms-11-01807-f007:**
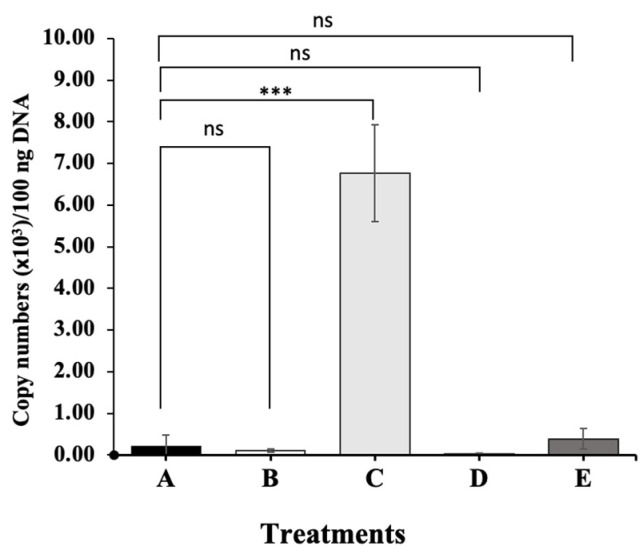
Quantitative real-time PCR analysis of the DNA copy number of the *AcgyrB* gene of the targeted probiotic *Acinetobacter* KU011TH in fish groups treated with the varied synbiotic dosages compared to the control group fish. A–F indicate control, chitosan, chitosan+ 1 × 10^8^, chitosan+ 1 × 10^9^ and chitosan+ 1 × 10^10^ CFU/kg groups, respectively. Asterisks *** on each graph are considered significant at *p* < 0.001 (*n* = 4), and ns indicates nonsignificant compared with control (*p* > 0.05).

**Figure 8 microorganisms-11-01807-f008:**
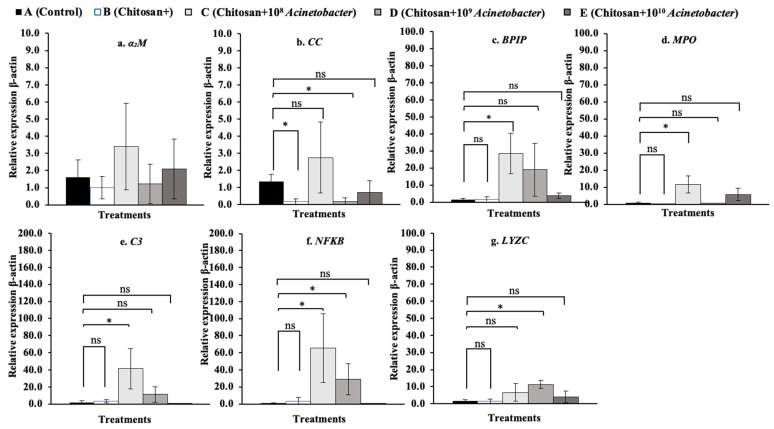
Relative expression profiles of immune-related genes in whole blood of hybrid catfish (*C. macrocephalus* × *C. gariepinus*) after feeding with symbiotic-supplemented feed in a long-term trial. Asterisks * on each graph are considered significant at *p* < 0.05, (*n* = 4), and ns indicates nonsignificant compared with control (*p* > 0.05).

**Figure 9 microorganisms-11-01807-f009:**
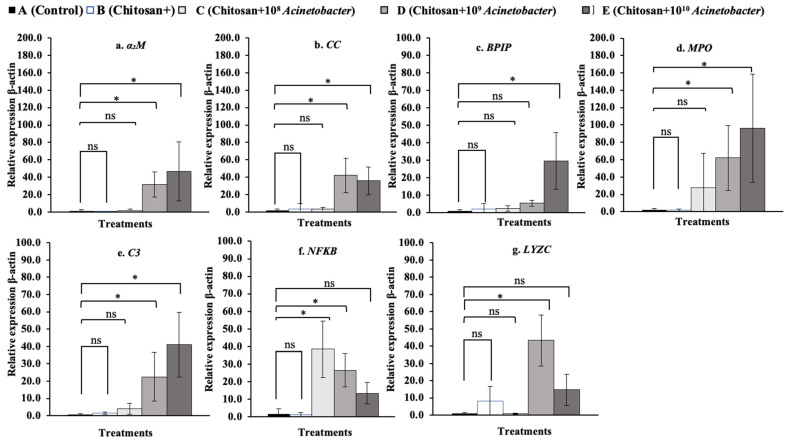
Relative expression profiles of immune-related genes in the head kidney of hybrid catfish (*C. macrocephalus* × *C. gariepinus*) after feeding with symbiotic-supplemented feed in a long-term trial. Asterisks * on each graph are considered significant at *p* < 0.05 (*n* = 4), and ns indicates nonsignificant compared with control (*p* > 0.05).

**Figure 10 microorganisms-11-01807-f010:**
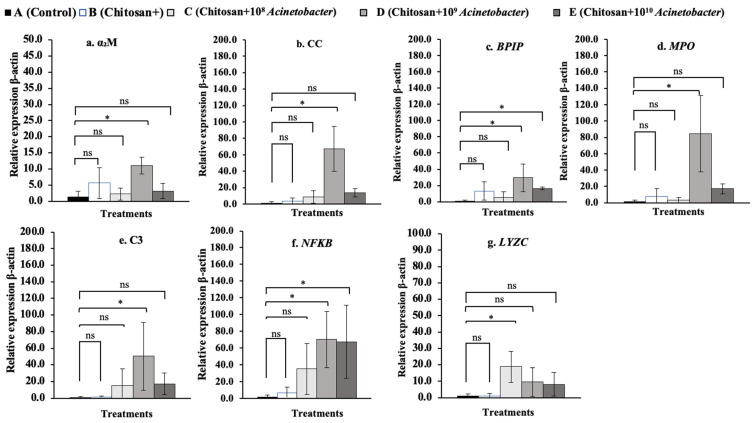
Relative expression profiles of immune-related genes in the spleen of hybrid catfish (*C. macrocephalus* × *C. gariepinus*) after feeding with symbiotic-supplemented feed in a long-term trial. Asterisks * on each graph are considered significant at *p* < 0.05 (*n* = 4), and ns indicates nonsignificant compared with control (*p* > 0.05).

**Figure 11 microorganisms-11-01807-f011:**
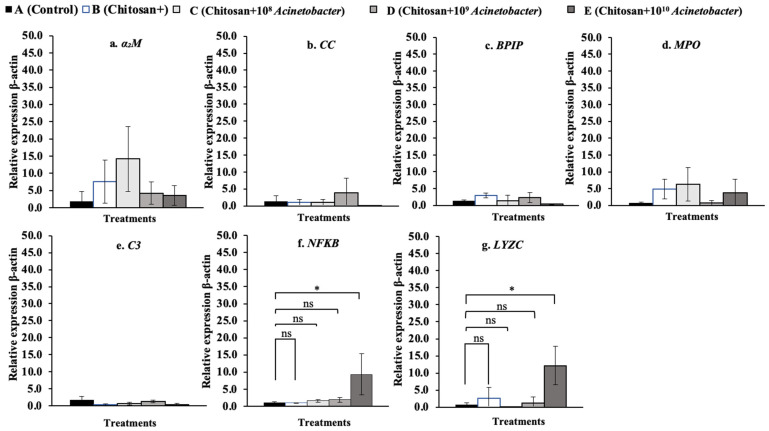
Relative expression profiles of immune-related genes in the liver of hybrid catfish (*C. macrocephalus × C. gariepinus*) after feeding with symbiotic-supplemented feed in a long-term trial. Asterisks * on each graph are considered significant at *p* < 0.05 (*n* = 4), and ns indicates nonsignificant compared with control (*p* > 0.05).

**Figure 12 microorganisms-11-01807-f012:**
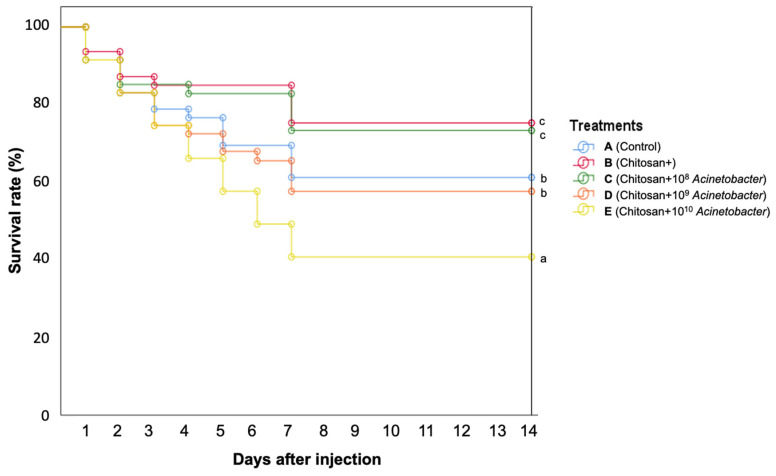
Survival analysis of fish fed prebiotic- and synbiotic-supplemented feed and challenged with *A. hydrophila* for 14 days (means ± SDs). Significant differences among tested groups are indicated by different letters (*p* < 0.05).

**Table 1 microorganisms-11-01807-t001:** Synbiotic effects of chitosan and synbiotic supplementation on growth performance of hybrid catfish. Each column represents different feeding treatments. All values are expressed as the means ± SDs obtained from five different groups (*n* = 30 or less). Data with different superscripts in the same row indicate significant differences compared with the control (*p* < 0.05). Treatments, A = control; B = chitosan; C, D, and E = symbiotic-supplemented groups with *Acinetobacter* KU011TH at 1 × 10^8^, 1 × 10^9^, and 1 × 10^10^ CFU/kg feed, respectively.

	Treatments				
Growth Parameters	A (Control)	B	C	D	E
Initial length (IL; cm)	8.87 ± 0.26	8.78 ±0.25	8.75 ± 0.20	9.03 ± 0.19	8.70 ± 0.10
Final length (FL; cm)	25.93 ± 0.61	26.28 ± 0.22	26.65 ± 0.72	26.28 ± 0.41	25.93 ± 0.50
Initial weight (IW; g)	5.49 ± 0.32	5.67 ± 0.48	5.72 ± 0.59	5.74 ± 0.31	5.90 ± 0.42
Final weight (FW; g)	150.63 ± 13.80 ^a^	165.28 ± 11.28 ^a^	156.80 ± 22.82 ^a^	155.03 ± 10.93 ^a^	128.67 ± 3.66 ^b^
Length gain (LG; cm)	17.06 ± 0.79	17.50 ± 0.34	17.91 ± 0.82	17.25 ± 0.50	17.23 ± 0.41
Weight gain (WG; g)	145.15 ± 18.01	159.61 ± 9.31	151.08 ± 27.84	149.29 ± 9.40	122.77 ± 6.27
Average daily gain (ADG; g/individual/day)	1.20 ± 0.14	1.31 ± 0.07	1.27 ± 0.19	1.23 ± 0.07	1.02 ± 0.05
Specific growth rate (SGR; %/day)	1.25 ± 0.05 ^a^	1.28 ± 0.06 ^a^	1.25 ± 0.04 ^a^	1.24 ± 0.03 ^a^	1.15 ± 0.03 ^b^
Feed conversion ratio (FCR)	1.81 ± 0.02	1.68 ± 0.30	1.82 ± 0.23	1.90 ± 0.11	1.98 ± 0.06
Accumulative mortality (%)	7.78 ± 5.09	7.78 ± 7.70	5.56 ± 6.94	6.67 ± 3.85	0.83 ± 1.67

## Data Availability

Data is available under request.

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
