# Peer review of "Long-Term Application of a Synbiotic Chitosan and Acinetobacter KU011TH Mixture on the Growth Performance, Health Status, and Disease Resistance of Hybrid Catfish (Clarias gariepinus × C. macrocephalus) during Winter"

_microorganisms, 2023, doi:10.3390/microorganisms11071807_

Round 1
Reviewer 1 Report
Major concern
1. Introduction and discussion are too long and out of focus. Delete Line 33-51. Also, most general statements/presentations in discussion part can be removed. For example, line 540-542, line 558-562, line 685-687, et al.
2. Data statistics and presentation are not very scientific and formal. For example, all figures and tables.
3. Several conclusions did not support by data, e. g. all growth parameters among the tested groups did not significantly differ from the control. But the weight in group E was significantly lower than the control form figure 1.
Minor concern
1. Figure 1c, the pattern of letter (indicating significant) at the time point of 9 week is not uniform with the others.
2. The data in Figure 1 and Table 1 partially overlapped?
3. Table 1, Why are the same letters marked between groups that are not significantly different?
none
Author Response
Reviewer 1# Major concern 1. Introduction and discussion are too long and out of focus. Delete Line 33-51. Also, most general statements/presentations in discussion part can be removed. For example, line 540-542, line 558-562, line 685-687, et al. Responses: Thank you so much for this suggestion. We have revised and removed these redundant contents of these paragraphs in our manuscript. 2. Data statistics and presentation are not very scientific and formal. For example, all figures and tables. Responses: Thank you so much for this crucial suggestion. We have tried to reanalyze statistical data and this information have been revised in the current version of our manuscript. 3. Several conclusions did not support by data, e. g. all growth parameters among the tested groups did not significantly differ from the control. But the weight in group E was significantly lower than the control form figure 1. Responses: Thank you so much for these key comments. These data have been carefully revised to properly support the critical parts suggested by the reviewer. Minor concerns 1. Figure 1c, the pattern of letter (indicating significant) at the time point of 9 week is not uniform with the others. Responses: Thank you so much for this comment. We have corrected this suggested datum. 2. The data in Figure 1 and Table 1 partially overlapped? Responses: Thank you so much for this suggestion. We well agree and understand this suggestion. In the Table 1, we had to indicate statistical analysis of weight at the final tested period that may little overlap with Figure 1C which demonstrated more details of weekly data. 3. Table 1, Why are the same letters marked between groups that are not significantly different? Responses: Thank you so much for this suggestion. We removed all the same letters from this table, where there were significant differences among all tested groups.

Reviewer 2 Report
the authors investigated the effect of long-term application of a synbiotic chitosan and acinetobacter ku011th mixture on the growth performance, health status and disease resistance in hybrid catfish (clarias gariepinus × c. macrocephalus) during winter. They designed six treatments to test their hypothesis. This manuscript (MS) was clearly written and easy to understand. They covered a wide range of physical factors, from growth to histology, still some gaps are here in terms of provided data. This work can help the sustainability of this species farming as few studies have been done on this topic. However, some major issues significantly compromised the quality of this MS.
First, the manuscript needs to be edited by a native English speaker to improve the language of the MS and fix errors.
Line 19-21 and other parts: the name of treatments are confusing and make the MS hard to read. Please change them to something more meaningful. For example, control, Prebio, Prebio+8, Prebio+9, and Prebio+10. Please update the MS, text and figures with this change.
· Line 23, temperature.
· Line 35-53, combine two paragraphs and delete irrelevant parts.
· In all figures, please set up the vertical y axis to ten numbers instead of five. It will help to read the exact number easier. Also, please select more distinctive colors for treatments in figures, like red, green blue, etc,
· In tables and figures, when there is no significant difference, please do not use any subset. The lack of a subset means there is no significant difference among treatments.
· In the figures ad the name of treatments that I mentioned earlier in x-axis.
· Figure 5, what the asterisk means in the figure?. Please add complete information in the legends of figures which analysis did you do and what the subsets mean.
· Figure 8-11, no need to report points in plot. Please change them to a normal column bar (like other figures) considering the consistency of colors for each treatment.
· Please make sure you defined the abbreviations for the first time in the MS. One time in the abstract and one time after the introduction if you use it in both the abstract and introduction.
· Here and elsewhere, report P uppercase and italic (P<0.05).
· Throughout the MS, if there is no significant difference, no need to report P-value.
· Please reorder the keywords alphabetically and capitalize each word.
· Please write the abstract more numerically about the results. You can do it by adding their numbers in parentheses.
· Here and throughout the MS, please first mention the common name plus scientific name, and for the rest of the MS, just report the common name.
· Please update the introduction with recent works as many studies are available from the last two years, which were not included in this section.
· Please mention the novelty of your work in the last paragraph of the introduction.
· For each analysis, please clarify how many fish were taken.
• As a general comment: please focus on fish as hips of references and studies are available, and no need to cite other vertebrates.
· Some parts of the discussion are better updated with research in 2022 and 2023 as they refer to some old references. Please update the discussion with the latest studies as much as possible.
· Although you wrote this section well, you can still improve it by answering these questions and annotating them into the discussion section. Why were these results observed? Discuss more possible reasons.
· The conclusion needs to be revised and more comprehensive concepts should be added there.
·
Tables and Figures
• Please explain a little bit about your experimental treatments, per each Table and Figure. Each Table and figure should represent enough information separately from the text.
• Double-check the units and titles of all Tables.
• Please mention in the footnote of all Tables which kind of statistical method you used for comparing the means.
When revising your manuscript, please consider all issues mentioned in the reviewers' comments carefully with clear outlines for every change made in response to their comments including suitable rebuttals for any comments you deem inappropriate. Please itemize your response to each review comment, and highlight the revised at re-submission.
Best regards
Not bad, but it needs some effort.
Author Response
Reviewer 2#
The authors investigated the effect of long-term application of a synbiotic chitosan and acinetobacter ku011th mixture on the growth performance, health status and disease resistance in hybrid catfish (Clarias gariepinus × C. macrocephalus) during winter. They designed six treatments to test their hypothesis. This manuscript (MS) was clearly written and easy to understand. They covered a wide range of physical factors, from growth to histology, still some gaps are here in terms of provided data. This work can help the sustainability of this species farming as few studies have been done on this topic. However, some major issues significantly compromised the quality of this MS.
First, the manuscript needs to be edited by a native English speaker to improve the language of the MS and fix errors.
Responses: Thank you so much for this suggestion. The current version of our manuscript was carefully polished by a native speaker who is keen in this field.
Line 19-21 and other parts: the name of treatments are confusing and make the MS hard to read. Please change them to something more meaningful. For example, control, Prebio, Prebio+8, Prebio+9, and Prebio+10. Please update the MS, text and figures with this change.
Responses: Thank you so much for this suggestion. We well agreed with this comment, however, to avoid using the term “prebiotics” (suggested by the editor and other reviewer), we have to remain the specific symbols A-E to indicate the used treatments in our manuscript.
- Line 35-53, combine two paragraphs and delete irrelevant parts.
Responses: Thank you so much for this suggestion. We have carefully revised this suggested part.
- In all figures, please set up the vertical y axis to ten numbers instead of five. It will help to read the exact number easier. Also, please select more distinctive colors for treatments in figures, like red, green blue, etc,
- In tables and figures, when there is no significant difference, please do not use any subset. The lack of a subset means there is no significant difference among treatments.
Responses: Thank you so much for this suggestion. These critical contents have been corrected.
- In the figures ad the name of treatments that I mentioned earlier in x-axis.
Responses: Thank you so much for this suggestion. We well agreed with this comment, however, to avoid using the term “prebiotics” (suggested by the editor and other reviewer), we have to remain the specific symbols A-E to indicate the used treatments in our manuscript.
- Figure 5, what the asterisk means in the figure?. Please add complete information in the legends of figures which analysis did you do and what the subsets mean.
Responses: Thank you so much for this suggestion. We have previously indicated what * or ** were used for in the figure legend of the Figure 2.
- Figure 8-11, no need to report points in plot. Please change them to a normal column bar (like other figures) considering the consistency of colors for each treatment.
- Please make sure you defined the abbreviations for the first time in the MS. One time in the abstract and one time after the introduction if you use it in both the abstract and introduction.
Responses: Thank you so much for this suggestion.
- Here and elsewhere, report P uppercase and italic (P<0.05).
Responses: Thank you so much for this suggestion. We have carefully checked and corrected these suggested points throughout the manuscript.
- Throughout the MS, if there is no significant difference, no need to report P-value.
Responses: Thank you so much for this suggestion. We have carefully checked and corrected these suggested points throughout the manuscript.
- Please reorder the keywords alphabetically and capitalize each word.
Responses: Thank you so much for this suggestion. We have corrected these suggested points in the last version of the manuscript.
- Please write the abstract more numerically about the results. You can do it by adding their numbers in parentheses.
Responses: Thank you so much for this suggestion. We have corrected these suggested points in the last version of the manuscript.
- Here and throughout the MS, please first mention the common name plus scientific name, and for the rest of the MS, just report the common name.
Responses: Thank you so much for this suggestion. We have corrected these suggested points in the last version of the manuscript.
- Please update the introduction with recent works as many studies are available from the last two years, which were not included in this section.
Responses: Thank you so much for this suggestion. We have corrected these suggested points in the last version of the manuscript.
- Please mention the novelty of your work in the last paragraph of the introduction.
Responses: Thank you so much for this suggestion. We have corrected these suggested points in the last version of the manuscript.
- For each analysis, please clarify how many fish were taken.
Responses: Thank you so much for this suggestion. We have corrected these suggested points in the last version of the manuscript.
- As a general comment: please focus on fish as hips of references and studies are available, and no need to cite other vertebrates.
Responses: Thank you so much for this suggestion. We have corrected these suggested points in the last version of the manuscript.
- Some parts of the discussion are better updated with research in 2022 and 2023 as they refer to some old references. Please update the discussion with the latest studies as much as possible.
Responses: Thank you so much for this suggestion. We have corrected these suggested points in the last version of the manuscript.
- Although you wrote this section well, you can still improve it by answering these questions and annotating them into the discussion section. Why were these results observed? Discuss more possible reasons.
Responses: Thank you so much for this suggestion. We have corrected these suggested points in the last version of the manuscript.
- The conclusion needs to be revised and more comprehensive concepts should be added there.
Responses: Thank you so much for this suggestion. We have corrected these suggested points in the last version of the manuscript.
Tables and Figures
- Please explain a little bit about your experimental treatments, per each Table and Figure. Each Table and figure should represent enough information separately from the text.
Responses: Thank you so much for this suggestion. We have corrected these suggested points in the last version of the manuscript.
- Double-check the units and titles of all Tables.
Responses: Thank you so much for this suggestion. We have corrected these suggested points in the last version of the manuscript.
- Please mention in the footnote of all Tables which kind of statistical method you used for comparing the means.
Responses: Thank you so much for this suggestion. We have corrected these suggested points in the last version of the manuscript.

Round 2
Reviewer 1 Report
the authors have solved my concerns
easy to understand
Author Response
Response to reviewer
First of all, we are really appreciating all comments suggesting on our manuscripts for the third round. These comments are very useful for improving quality of our scientific research to warrant an acceptable research document eventually.
In the current version of our manuscript, we carefully corrected it in all concerns by strictly following the valuable suggestions of the anonymous reviewers, indicating as yellow highlights in the manuscript.
We appreciate your times and look forward to having your response soon.
Sincerely
Prapansak Srisapoome, Ph. D.
Department of Aquaculture
Faculty of Fisheries, Kasetsart University
Bangkok, Thailand 10900.
